# Unsupervised Scalable Representation Learning for Multivariate Time Series

**Jean-Yves Franceschi,**[*1] **Aymeric Dieuleveut**[2] **& Martin Jaggi**[2]
[1]Sorbonne Université, CNRS, Laboratoire d'informatique de Paris 6, LIP6, F-75005 Paris, France
[2]MLO, EPFL, Lausanne CH-1015, Switzerland
`jean-yves.franceschi@lip6.fr`,
`{aymeric.dieuleveut,martin.jaggi}@epfl.ch`

## Abstract

Time series constitute a challenging data type for machine learning algorithms, due to their highly variable lengths and sparse labeling in practice. In this paper, we tackle this challenge by proposing an unsupervised method to learn universal embeddings of time series. Unlike previous works, it is scalable with respect to their length and we demonstrate the quality, transferability and practicability of the learned representations with thorough experiments and comparisons. To this end, we combine an encoder based on causal dilated convolutions with a novel triplet loss employing time-based negative sampling, obtaining general-purpose representations for variable length and multivariate time series.

## 1 Introduction

We investigate in this work the topic of unsupervised general-purpose representation learning for time series. In spite of the increasing amount of work about representation learning in fields like natural language processing or videos, few articles explicitly deal with general-purpose representation learning for time series without structural assumption on non-temporal data. This problem is indeed challenging for various reasons. First, real-life time series are *rarely or sparsely labeled*. Therefore, *unsupervised* representation learning would be strongly preferred. Secondly, methods need to deliver compatible representations while allowing the input time series to have unequal lengths. Thirdly, scalability and efficiency both at training and inference time is crucial, in the sense that the techniques must work for both short and long time series encountered in practice.

Unlike prior work (Hyvarinen & Morioka, 2016; Lei et al., 2017; Malhotra et al., 2017; Wu et al., 2018), we tackle all these challenges by proposing an *unsupervised* method to learn *general-purpose representations* for *multivariate* time series that comply with the issues of *varying and potentially high lengths* of the studied time series. We finally assess the quality of the learned representations on standard datasets to ensure their universality. In particular, we show that our representations obtain *competitive performance* on classification tasks, while showing *transferability* properties. We also evaluate our representations on a real-life dataset including very long time series, on which we demonstrate *scalability* and generalization ability *across different tasks* beyond classification.

## 2 Unsupervised Training

We seek to train an encoder network without having to jointly train a decoder in an autoencoder framework like standard representations learning methods, as done by Malhotra et al. (2017), since it would induce a larger computational cost. To this end, we choose to use a novel triplet loss, inspired by an approach used for word representation learning with word2vec (Mikolov et al., 2013). As far as we know, this work is the first in the general time series literature (with no focus on specific data structures like videos or audio) to propose a triplet loss for feature learning, and especially one handling time series of different lengths.

---

[*]Work partially done while studying at ENS de Lyon and MLO, EPFL.

The objective is to ensure that similar time series obtain similar representations, with no supervision to learn such similarity. The assumption made in the CBOW model of word2vec is twofold. The representation of the *context* of a word should probably be, on one hand, close to the one of this word (Goldberg & Levy, 2014), and, on the other hand, distant from the one of randomly chosen words, since they are probably unrelated to the original word's context. The corresponding loss then pushes pairs of (context, word) and (context, random word) to be linearly separable. This is called *negative sampling*.

To adapt this principle to time series, we consider a random subseries[1] $x^{\text{ref}}$ of a given time series $y_i$. Then, on one hand, its representation should probably be close to the one of any of its own subseries $x^{\text{pos}}$ (a *positive* example). On the other hand, if we consider another subseries $x^{\text{neg}}$ (a *negative* example) chosen at random (in a different random time series $y_j$ if a dataset is available, or in the same time series if it is long enough and not stationary), then its representation should probably be distant from the one of $x^{\text{ref}}$. See Figure 1 for an illustration. Following the

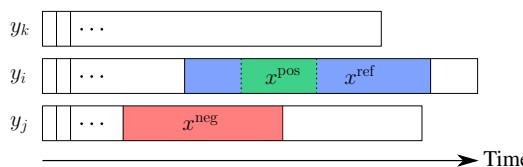

Figure 1: Choices of $x^{\text{ref}}$, $x^{\text{pos}}$ and $x^{\text{neg}}$.

comparison with word2vec, $x^{\text{pos}}$ corresponds to a word, $x^{\text{ref}}$ to its context, and $x^{\text{neg}}$ to a random word. To improve the stability and convergence of the training procedure as well as the experimental results of our learned representations, we introduce, as in word2vec, several negative samples $(x_k^{\text{neg}})_{k \in [\![1,K]\!]}$, chosen independently at random. The objective to be minimized corresponding to these choices, similarly to the one of word2vec with its shallow network replaced by a deep encoder network $f(.,\theta)$ with parameters $\theta$, is:

$$-\log\left(\sigma\left(f\left(x^{\text{ref}},\theta\right)^\top f\left(x^{\text{pos}},\theta\right)\right)\right) - \sum_{k=1}^{K}\log\left(\sigma\left(-f\left(x^{\text{ref}},\theta\right)^\top f\left(x_k^{\text{neg}},\theta\right)\right)\right),$$

where $\sigma$ is the sigmoid function. This loss pushes the computed representations to distinguish between $x^{\text{ref}}$ and $x_k^{\text{neg}}$, and to assimilate $x^{\text{ref}}$ and $x^{\text{pos}}$. Note that $f$, which we choose to be the network presented in Section 3, must output fixed-length representations for all time series of all possible input sizes. Overall, the training procedure consists in traveling through the training dataset for several epochs (possibly using mini-batches), picking tuples $\left(x^{\text{ref}}, x^{\text{pos}}, \left(x_k^{\text{neg}}\right)_k\right)$ at random as detailed in Section A, and performing a minimization step on the corresponding loss for each pair.

This training procedure is interesting in that it is efficient enough to be run over long time series (see Section 4) with an efficient encoder (see Section 3), thanks to its decoder-less design and the separability of the loss, on which a backpropagation per term can be performed to save memory.

## 3   ENCODER ARCHITECTURE

We choose to use deep convolutional neural networks with *dilated* convolutions to handle time series. Popularized for sequence generation (Oord et al., 2016), dilated convolutional networks have never been used for unsupervised representation learning, to our knowledge. Compared to recurrent neural networks, which are inherently designed for sequence-modeling tasks and thus sequential, these networks are efficient as they are highly parallelizable on modern hardware such as GPUs. Moreover, using dilated convolutions rather than full convolutions allows to better capture long-range dependencies at constant depth by exponentially increasing the receptive field of the network (Oord et al., 2016; Yu & Koltun, 2016; Bai et al., 2018).

Our model consists of stacks of dilated *causal* convolutions (see Figure 2a), which map a sequence to a sequence of the same length, such that the $i$-th element of the output sequence is computed using only values up until the $i$-th element of the input sequence, for all $i$. It is thus called causal, since the output value corresponding to a given time step is not computed using future input values. Causal convolutions allow to organize the computational graph so that, in order to update its output of a time series when an element is added at its end, one only has to evaluate the graph shown in Figure 2a rather than the full graph. This may save some computational time during testing.

---

[1]I.e., a subsequence of a time series composed by consecutive time steps of this time series.

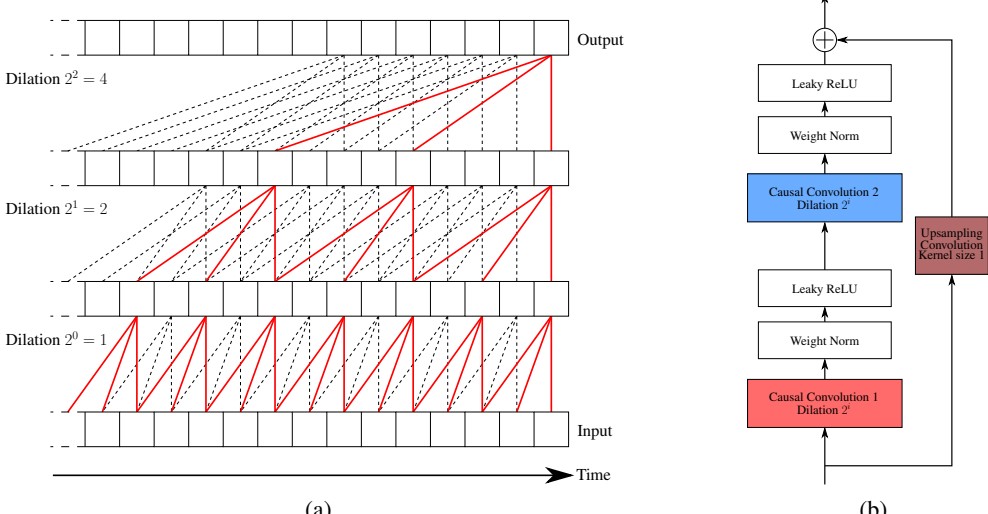

Figure 2: (a) Illustration of three stacked dilated causal convolutions. Lines between each sequence represent their computational graph. Red solid lines highlight the dependency graph for the computation of the last value of the output sequence, showing that no future value of the input time series is used to compute it. (b) Composition of the $i$-th layer of the chosen architecture.

Inspired by Bai et al. (2018), we build each layer of our network as a combination of causal convolutions, weight normalizations (Salimans & Kingma, 2016), leaky ReLUs and residual connections (see Figure 2b). Each of these layers is given an exponentially increasing dilation parameter ($2^i$ for the $i$-th layer). The output of this causal network is then given to a global softmax pooling layer squeezing the temporal dimension and aggregating all temporal information in a fixed-size vector (Wang et al., 2017). The output of the encoder is then a linear transformation of the latter vector.

## 4 EXPERIMENTAL RESULTS

We present in this section the experiments conducted to assess the quality of our representations. Some additional results, as well as the full training process and hyperparameter choices, are detailed in Sections B, C and D. Code corresponding to these experiments is publicly available.[2]

**Classification.** We first assess the *quality* of our learned representations in a standard manner by using them for time series classification. We show that our method outperforms the state-of-the-art unsupervised methods, and notably achieves performance close to the supervised state-of-the-art.

For each considered dataset with a train / test split, we unsupervisedly train an encoder using its train set. We then train an SVM with radial basis function kernel on top of the learned features using the train labels of the dataset, similarly to concurrent works TimeNet (Malhotra et al., 2017) and RWS (Wu et al., 2018), and output the corresponding classification score on the test set. As $K$ can have a significant impact on the performance, we present a *combined* version of our method, where pairs of encoders and SVMs for different values of $K$ (see Section D) are combined in a voting classifier.

We choose to compare ourselves to the four best state-of-the-art classifiers of univariate time series studied by Bagnall et al. (2017): COTE – replaced by its improved version HIVE-COTE (Lines et al., 2018) –, ST (Bostrom & Bagnall, 2015), BOSS (Schäfer, 2015) and EE (Lines & Bagnall, 2015), on the first 85 datasets of the UCR archive (Dau et al., 2018).[3] We also add DTW (which is a one-nearest neighbor classifier with Dynamic Time Warping as measure) as a baseline to the comparison. Note that almost half of the tested datasets have *few available labels* as their train set is of size at most 200.

---

[2]https://github.com/White-Link/UnsupervisedScalableRepresentationLearningTimeSeries.

[3]The new UCR archive includes 43 new datasets on which no results other than ours have been produced yet.

We observe (see Figure 3, as well as Figures 4 and 5 in the appendices) that our method is globally second-to-best (with average rank 2.94), only beaten by HIVE-COTE (1.67) and equivalent to ST (2.95). HIVE-COTE being a powerful ensemble method containing, among others, all classifiers in this comparison and thus difficult to outperform, we highlight that our method based on unsupervised representations achieves remarkable performance as it *matches the second-to-best studied supervised method*, and in particular is *at the level of the best performing method included in HIVE-COTE*. Besides their performance, our representations are also transferable, as explained in Section B. We highlight that this performance is achieved, when the encoder is trained, *efficiently* both in terms of time (training is a matter of seconds) and space (only a simple SVM must be trained). Partial results (taken as reported in the original articles; see Section E) also indicate that our method *consistently outperforms both concurrent unsupervised methods*

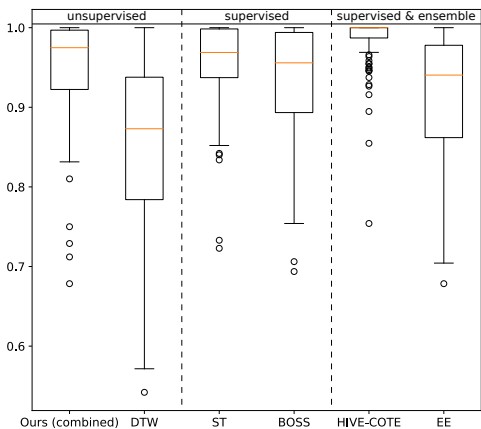

Figure 3: Boxplot of the ratio of the accuracy versus maximum achieved accuracy of compared methods on the first 85 UCR datasets.

TimeNet and RWS (on, respectively, 13 and 9 out of 13 and 12 UCR datasets). They also indicate that our method is beaten (on $68\%$ out of 44 UCR datasets) by FCN (Wang et al., 2017), which are neural networks trained for supervised classification, and thus expected to beat our neural network trained unsupervisedly.

**Regression.** We show the *applicability* and *scalability* of our method on *long* time series *without labeling* for regression tasks on the Individual Household Electric Power Consumption (IHEPC) dataset from the UCI Machine Learning Repository (Dheeru & Karra Taniskidou, 2017), which is a single time series of length $2\,075\,259$ monitoring the minute-averaged electricity consumption of a single household in France for four years. We split this time series into train (first $5 \times 10^5$ measurements, approximately a year) and test (remaining measurements), and normalize it to a zero-mean and unit variance time series. The encoder is trained over the train time series on a single Nvidia Tesla P100 GPU in no more than a few hours, showing that our training procedure is *scalable* to long time series.

We consider the tasks of, given a previous time windows of a day ($1\,440$ measurements) and a quarter ($12 \cdot 7 \cdot 1\,440$ measurements), predicting the evolution of the mean value of the series during the next day (respectively, quarter). We compare linear regressors applied on the raw time series and on the corresponding representations. Results and execution times on a Nvidia Titan Xp GPU are presented in Table 1.[4] On both scales of inputs, our representations induce only a slightly degraded performance but provide a large efficiency improvement.

Table 1: Results obtained on the IHEPC dataset.

| Task | Metric | Representations | Raw values |
|---|---|---|---|
| Day | Test MSE | $\mathbf{8.92 \cdot 10^{-2}}$ | $\mathbf{8.92 \cdot 10^{-2}}$ |
| | Wall time | **12s** | 3min 1s |
| Quarter | Test MSE | $7.26 \cdot 10^{-2}$ | $\mathbf{6.26 \cdot 10^{-2}}$ |
| | Wall time | **9s** | 1h 40min 15s |

## 5 CONCLUSION

We presented an unsupervised general-purpose representation learning method for time series that is scalable and produces high-quality and easy-to-use embeddings. They are generated by an encoder formed by dilated convolutions that admits variable-length inputs, and trained with a triplet loss using novel negative sampling for time series. Conducted experiments show that these representations are universal and can efficiently be used for classification and regression with few or no available labels.

---

[4]While acting on representations of the same size, the quarterly linear regressor is slightly faster than the daily one because the number of quarters in the considered time series is smaller than the number of days.

ACKNOWLEDGMENTS

We would like to acknowledge Patrick Gallinari, Sylvain Lamprier, Mehdi Lamrayah, Sidak Pal Singh, Andreas Hug, Jean-Baptiste Cordonnier and François Fleuret for helpful comments and discussions at various stages of this project, as well as all contributors to the datasets and archives we used for this project (Dau et al., 2018; Bagnall et al., 2018; Dheeru & Karra Taniskidou, 2017).

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

# APPENDICES

## A    DETAILED CHOICES OF POSITIVE AND NEGATIVE SAMPLES

In practice, we pick tuples $\left(x^{\text{ref}}, x^{\text{pos}}, (x_k^{\text{neg}})_{k \in [\![1,K]\!]}\right)$ in the following manner. We iterate over the available dataset for a given number of epochs (given as hyperparameter). For each train time series $z$, the length of $x^{\text{pos}}$ is chosen uniformly at random in $[\![1, \text{size}(z)]\!]$; then the size of $x^{\text{ref}}$ is chosen uniformly at random in $[\![\text{size}(x^{\text{pos}}), \text{size}(z)]\!]$, and $x^{\text{ref}}$ is chosen uniformly at random among all subseries of $z$ of the chosen size. Similarly, $x^{\text{pos}}$ is chosen uniformly at random in $x^{\text{ref}}$. The choice of $(x_k^{\text{neg}})_{k \in [\![1,K]\!]}$ consists in simply choosing uniformly at random the time series which they will be drawn from, then their length, then picking them at random as well according to those chosen parameters. The generalization of this procedure to mini-batch training is straightforward, so we do not detail it.

The length of the negative examples can either be the same for all samples and equal to $\text{size}(x^{\text{pos}})$, or be chosen at random similarly to $\text{size}(x^{\text{pos}})$. The first case is suitable when all time series in the dataset have equal lengths, and speeds up the training procedure thanks to computation factorizations; the second case is only used when time series in the dataset do not have the same lengths, as we saw no other difference than time efficiency between the two cases in our experiments.

One might consider capping the length of $x^{\text{neg}}$ in the case where only a single long time series is available, so that it does not contain $x^{\text{ref}}$, but this only happens with low probability and does not harm the performance of the learned representations in our experiments.

## B    FURTHER CLASSIFICATION RESULTS

**Full scores.**    We report in Table 2 scores for some UCR datasets as well as full scores for our method on the first 85 UCR datasets in Table 3. Full scores for DTW, ST, BOSS, HIVE-COTE and EE are reported online by Bagnall et al. (2017).[5]

**Transferability.**    We include in Table 2 the classification accuracy for each dataset of an SVM trained on this dataset using the representations computed by an encoder, which was trained *on another dataset* (FordA, with $K = 5$), to test the transferability of our representations.

We observe that the scores achieved by this SVM trained on transferred representations are close to the scores reported when the encoder is trained on the same dataset as the SVM, showing the *transferability* of our representations from a dataset to another, and from time series to other time series *with different lengths*.

**Multivariate Time Series Classification.**    We tested our method on all 30 datasets of the newly released UEA archive (Bagnall et al., 2018). For each dataset, each dimension of the time series was preprocessed independently from the other dimensions by normalizing its mean and variance.

The UEA archive has been designed as a first attempt to provide a standard archive for multivariate time series classification such as the UCR one for univariate series. As it has only been released recently, we could not compare our method to state-of-the-art classifiers for multivariate time series. However, we provide a comparison with DTW$_{\text{D}}$ as baseline using results provided by Bagnall et al. (2018). DTW$_{\text{D}}$ (dimension-Dependent DTW) is a possible extension of DTW in the multivariate setting, and is the best baseline studied by Bagnall et al. (2018).

Overall, our method matches or outperforms DTW$_{\text{D}}$ on $72\%$ of the UEA datasets, which indicates a good performance. As this archive is destined to grow and evolve in the future, and without further comparisons, no additional conclusion can be drawn.

---

[5] http://www.timeseriesclassification.com/singleTrainTest.csv.

Table 2: Accuracy scores of variants of our method compared with those of DTW (unsupervised), ST and BOSS (supervised) and HIVE-COTE and EE (supervised ensemble methods), on some UCR datasets. Bold scores indicate the best performing method.

| Dataset | Ours (unsupervised) | | Unsup. | Supervised | | Supervised ensemble | |
| --- | --- | --- | --- | --- | --- | --- | --- |
| | Combined | FordA ($K = 5$) | DTW | ST | BOSS | HIVE-COTE | EE |
| DiatomSizeReduction | 0.99 | 0.958 | 0.967 | 0.925 | 0.931 | 0.941 | 0.944 |
| ECGFiveDays | **1** | 0.768 | **1** | 0.984 | **1** | **1** | 0.82 |
| FordB | 0.798 | 0.764 | 0.62 | 0.807 | 0.711 | **0.823** | 0.662 |
| Ham | 0.657 | 0.723 | 0.467 | 0.686 | 0.667 | 0.667 | 0.571 |
| Phoneme | 0.272 | 0.225 | 0.228 | 0.321 | 0.265 | **0.382** | 0.305 |
| SwedishLeaf | 0.939 | 0.909 | 0.792 | 0.928 | 0.922 | **0.954** | 0.915 |

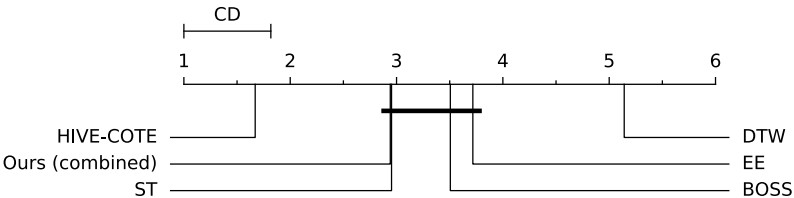

Figure 4: Critical difference diagram of the average ranks of the compared classifiers for the Nemenyi test, obtained with Orange (Demšar et al., 2013).

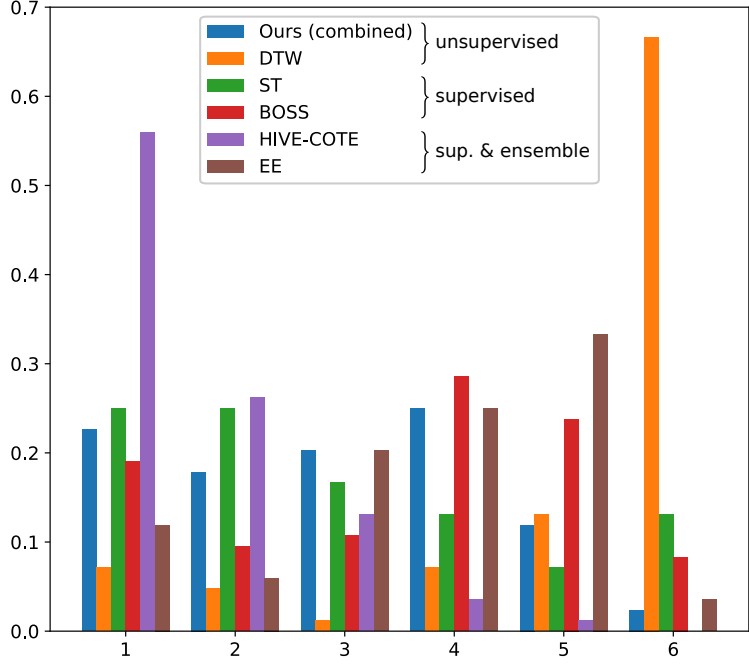

Figure 5: Distribution of ranks of compared methods for the first 85 UCR datasets.

## C  DETAILED TRAINING FOR CLASSIFICATION TASKS

We used Python 3 for implementation, with PyTorch 0.4.1 (Paszke et al., 2017) for neural networks and scikit-learn (Pedregosa et al., 2011) for SVMs. Each encoder was trained on a single Nvidia Titan Xp GPU with CUDA 9.0.

### C.1  SVM TRAINING AND EARLY STOPPING

We perform no hyperparameter optimization on the architecture of our encoder, nor on the batch size or optimizer we use. We thus perform a single training procedure for each dataset and parameter $K$. The only parameters we dynamically tune are the number of epochs to train the encoder through an early stopping heuristic (stop training after a given number of epochs have been done without increasing a performance score and until a given number of epochs is reached, and keep the encoder corresponding to the best score), and the penalty $C$ of the error term of the SVM. Note that this tuning is done *without using test labels*.

Early stopping is only introduced in order to avoid, unlike for instance concurrent unsupervised work TimeNet (Malhotra et al., 2017), optimizing hyperparameters for each dataset, as they are substantially different from one another, and we stress that its introduction *does not introduce any bias compared to a fully unsupervised setting*. We further discuss in Section C.2 the use of early stopping and its impact on an unsupervised training. In particular, it is strictly equivalent to a more computationally demanding and fully unsupervised training, and remains optional.

In order to tune the error term of the SVM and monitor a performance test which is not the train classification score for the early stopping criterion, we use as performance score a cross-validation score on the training set in the following manner. To choose a penalty for the SVM, we freeze the encoder and pick the penalty that achieves the best cross-validation score on the representations of the train set. The early stopping criterion is then the cross-validation score of the best found penalty term. Note that if the train set or the number of training samples per class are too small, we do not use early stopping and choose a penalty $C = \infty$ for the SVM (which corresponds to no regularization).

This complex scheme is required because the avalaible UCR and UEA archives do not provide any additional validation set. Because lots of datasets are small, and to guarantee a fair comparison with concurrent methods which do not use any validation set, we designed the early stopping strategy to only use training labels.

### C.2  EARLY STOPPING DISCUSSION

With such an early stopping criterion, the entire method is then not fully unsupervised, because the labels are used to decide when to stop the learning procedure. This choice was mainly made to avoid having extra hyper-parameters to tune, and to save time on computations by avoiding a long training on some datasets with a small batch size. Indeed, datasets which we test our method on are varied and their size is particularly important with respect to training performance. As we keep the batch size and neural network hyperparameters constant across all datasets, dynamically tuning the number of epochs is useful in order to avoid overfitting on some datasets and save computational time on some others.

We highlight that *it does not change much the overall results*, as it improves the accuracy on some datasets, but worsens them on some others. As an example, we provide in Figure 6 the evolution of the test accuracy with respect to the number of epochs, showing that the stopping time is not optimal. Besides, the encoder can always be trained without label information (stopping after a certain number of epochs), or with very sparsely labeled time series. Comparison with other concurrent unsupervised methods is also fair as they can perform hyperparameter optimization, like TimeNet (Malhotra et al., 2017).

Moreover, this encoder training augmented with such an early stopping criterion is strictly equivalent to a more computationally demanding and *fully unsupervised* training. Indeed, consider the training of the encoder for a number of epochs equal to the maximal number of epochs under the early stopping procedure. If one records the weights of the encoder at the end of each epoch, then one could simulate the online early stopping heuristic in an offline fashion by iteratively computing the early stopping performance score, stopping when the early stopping conditions are met and retain

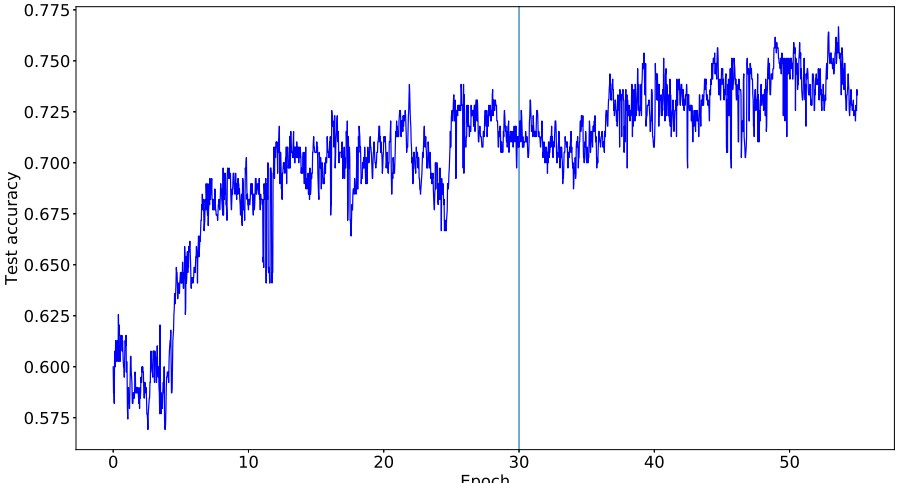

Figure 6: Evolution of the test accuracy during the training of the representation on the CricketX dataset from the UCR archive (with $K = 10$), with respect to the number of completed epochs. The test labels were only used for monitoring purposes and the test accuracy was computed after each mini-batch optimization. The vertical line marks the epoch selected by the early stopping heuristic. Test accuracy clearly increases during training, and the early stopping heuristic is suboptimal on this dataset.

the best set of weights. This way, the encoder training is fully unsupervised at the cost of longer and more complex training using the train labels. Note that exploratory experiments indicate that selecting the best performing set of weights over the whole number of epochs, instead of simulating early stopping, tends to give results similar to the ones obtained with early stopping.

## D  HYPERPARAMETERS

### D.1  DISCUSSION OF THE INFLUENCE OF $K$

As mentioned in Section 4, $K$ can have a significant impact on the performance of the encoder. We notably observed that $K = 1$ leads to statistically significantly lower scores compared to scores obtained when trained with $K > 1$ on the UCR datasets, jutifying the use of several negative examples during training. We did not observe any clear statistical difference between other values of $K$ on the whole archive; however, we nocited important differences between different values of $K$ when studying individual datasets. Therefore, we chose to include several pairs of encoders and SVMs in a voting classifier to avoid selecting $K$ as a hyperparameter.

### D.2  DETAILED CHOICES OF HYPERPARAMETERS

We train our models with the following parameters for time series classification:

- optimizer: Adam (Kingma & Ba, 2015) with learning rate $\alpha = 0.001$ and decay rates $\beta = (0.9, 0.999)$;
- SVM: penalty $C \in \left\{ 10^i \mid i \in [\![-4, 4]\!] \right\} \cup \{\infty\}$;
- encoder training:
  - number of negative samples: $K \in \{1, 2, 5, 10\}$ for univariate time series, $K \in \{5, 10, 20\}$ for multivariate ones;
  - batch size: 10;

- – maximum number of epochs: 400;
- – number of epochs to wait without performance improvement for early stopping: 25;
- architecture:
  - – number of channels in the intermediary layers of the causal network: 40;
  - – number of layers (depth of the causal network): 10,
  - – kernel size of all convolutions: 3;
  - – negative slope of the leaky ReLU: 0.01;
  - – number of output channels of the causal network (before max pooling): 320;
  - – dimension of the representations: 160.

For the Individual Household Electric Power Consumption dataset, changes are the following:

- number of negative samples: $K = 10$;
- batch size: 1;
- no early stopping;
- number of channels in the intermediary layers of the causal network: 30;
- number of output channels of the causal network (before max pooling): 160;
- dimension of the representations: 80.

# E    COMPARISON WITH FCN, TIMENET AND RWS

Comparisons with FCN, TimeNet and RWS are shown in Table 3.

Table 3: Accuracy scores of the combined version of our method compared with those of FCN (supervised), TimeNet and RWS (unsupervised), when available. Bold scores indicate the best performing method. 'X's indicate that a score were reported in the original paper, but was either obtained using transferability or on a reversed train / test split of the dataset, thus not comparable to other results for this dataset.

| Dataset | Ours (unsupervised) | Supervised | Unsupervised | |
| | Combined | FCN | TimeNet | RWS |
|---|---|---|---|---|
| Adiac | 0.775 | **0.857** | 0.565 | - |
| ArrowHead | **0.851** | - | - | - |
| Beef | 0.633 | **0.75** | - | 0.733 |
| BeetleFly | **0.85** | - | - | - |
| BirdChicken | **0.9** | - | - | - |
| Car | **0.817** | - | - | - |
| CBF | 0.998 | **1** | - | - |
| ChlorineConcentration | 0.75 | **0.843** | 0.723 | 0.572 |
| CinCECGtorso | 0.726 | **0.813** | - | - |
| Coffee | **1** | **1** | - | - |
| Computers | **0.704** | - | - | - |
| CricketX | 0.79 | **0.815** | 0.659 | - |
| CricketY | 0.726 | **0.792** | X | - |
| CricketZ | 0.767 | **0.813** | X | - |
| DiatomSizeReduction | **0.99** | 0.93 | - | - |
| DistalPhalanxOutlineCorrect | **0.75** | - | X | - |
| DistalPhalanxOutlineAgeGroup | **0.748** | - | X | - |
| DistalPhalanxTW | **0.676** | - | X | X |
| Earthquakes | **0.748** | - | - | - |
| ECG200 | **0.87** | - | - | - |
| ECG5000 | **0.94** | - | 0.934 | 0.933 |
| ECGFiveDays | **1** | 0.985 | X | - |
| ElectricDevices | **0.73** | - | 0.665 | - |
| FaceAll | 0.786 | **0.929** | - | - |
| FaceFour | 0.875 | **0.932** | - | - |
| FacesUCR | 0.893 | **0.948** | - | - |
| FiftyWords | **0.778** | 0.679 | - | - |
| Fish | 0.897 | **0.971** | - | - |
| FordA | **0.932** | - | X | - |
| FordB | **0.798** | - | X | X |
| GunPoint | 0.989 | **1** | - | - |
| Ham | **0.657** | - | - | - |
| HandOutlines | **0.922** | - | - | 0.843 |
| Haptics | 0.506 | **0.551** | - | - |
| Herring | **0.609** | - | - | - |
| InlineSkate | **0.418** | 0.411 | - | - |
| InsectWingbeatSound | 0.603 | - | - | **0.619** |
| ItalyPowerDemand | 0.937 | **0.97** | - | 0.969 |
| LargeKitchenAppliances | **0.861** | - | - | 0.792 |
| Lightning2 | 0.77 | **0.803** | - | - |
| Lightning7 | 0.808 | **0.863** | - | - |
| Mallat | 0.962 | **0.98** | - | 0.937 |
| Meat | **0.917** | - | - | - |
| MedicalImages | 0.778 | **0.792** | 0.753 | - |
| MiddlePhalanxOutlineCorrect | **0.838** | - | X | X |
| MiddlePhalanxOutlineAgeGroup | **0.636** | - | X | - |
| MiddlePhalanxTW | **0.578** | - | X | - |
| MoteStrain | 0.859 | **0.95** | - | - |
| NonInvasiveFatalECGThorax1 | 0.938 | **0.961** | - | 0.907 |
| NonInvasiveFatalECGThorax2 | 0.945 | **0.955** | - | - |

Table 3: Accuracy scores of the combined version of our method compared with those of FCN (supervised), TimeNet and RWS (unsupervised), when available. Bold scores indicate the best performing method. 'X's indicate that a score were reported in the original paper, but was either obtained using transferability or on a reversed train / test split of the dataset, thus not comparable to other results for this dataset.

| Dataset | Ours (unsupervised) | Supervised | Unsupervised | |
| | Combined | FCN | TimeNet | RWS |
| --- | --- | --- | --- | --- |
| OliveOil | **0.9** | 0.833 | - | - |
| OSULeaf | 0.814 | **0.988** | - | - |
| PhalangesOutlinesCorrect | **0.808** | - | 0.772 | - |
| Phoneme | **0.272** | - | - | - |
| Plane | **1** | - | - | - |
| ProximalPhalanxOutlineCorrect | **0.89** | - | X | 0.711 |
| ProximalPhalanxOutlineAgeGroup | **0.839** | - | X | X |
| ProximalPhalanxTW | **0.81** | - | X | - |
| RefrigerationDevices | **0.515** | - | - | - |
| ScreenType | **0.491** | - | - | - |
| ShapeletSim | **0.75** | - | - | - |
| ShapesAll | **0.865** | - | - | - |
| SmallKitchenAppliances | **0.717** | - | - | - |
| SonyAIBORobotSurface1 | 0.895 | **0.968** | - | - |
| SonyAIBORobotSurface2 | 0.94 | **0.962** | - | - |
| StarlightCurves | 0.964 | **0.967** | - | - |
| Strawberry | **0.957** | - | 0.93 | - |
| SwedishLeaf | 0.936 | **0.966** | 0.901 | - |
| Symbols | 0.954 | **0.962** | - | - |
| SyntheticControl | 0.987 | **0.99** | 0.983 | - |
| ToeSegmentation1 | **0.947** | - | - | - |
| ToeSegmentation2 | **0.908** | - | - | - |
| Trace | **1** | **1** | - | - |
| TwoLeadECG | 0.989 | **1** | - | - |
| TwoPatterns | **1** | 0.897 | 0.999 | 0.999 |
| UWaveGestureLibraryX | **0.81** | 0.754 | - | - |
| UWaveGestureLibraryY | **0.741** | 0.725 | - | - |
| UWaveGestureLibraryZ | **0.766** | 0.729 | - | - |
| UWaveGestureLibraryAll | **0.927** | - | - | - |
| Wafer | 0.996 | **0.997** | 0.994 | 0.993 |
| Wine | **0.815** | - | - | - |
| WordSynonyms | **0.691** | 0.58 | - | - |
| Worms | **0.74** | - | - | - |
| WormsTwoClass | **0.74** | - | - | - |
| Yoga | **0.87** | 0.845 | 0.866 | - |

