# OpenReview forum: "Unsupervised Scalable Representation Learning for Multivariate Time Series"
_ICLR.cc/2019/Workshop/LLD — LLD 2019_

### Official Review · AnonReviewer1 · 2019-04-06
**Exciting work in a challenging and useful unsupervised setting**

**Rating:** 5
**Confidence:** 2

**Review:**

This paper provides an unsupervised representation learning algorithm for performing classification and regression in multivariate time series. It relies on a combination of cutting-edge techniques: triplet loss, stacked causal dilated convolutions (à la WaveNet), weight normalization, and residual connections. Although these techniques had been published before in isolation, they had never been implemented in combination up to this paper. Therefore, the contributions of this paper are novel enough for the ICLR LLD workshop.

The discussion of prior literature is solid. However, i will point out that the claim
"this works is the first in the time series literature to propose a triplet loss for feature learning"
is wrong. The paper of Jansen et al. ICASSP 2017 "Unsupervised learning of semantic audio representations" (https://arxiv.org/abs/1711.02209) is one counterexample.

The rest of the paper is very clear and eloquent. I recommend this paper for acceptance.

---

### Official Review · AnonReviewer2 · 2019-04-07
**Weak reject**

**Rating:** 2
**Confidence:** 2

**Review:**

Paper summary:

This paper proposes a novel unsupervised embedding for time-series. Its architecture mainly consists in a series of dilated causal convolutions, followed by a temporal averaging to obtain a representation which is independent on the length of the time-series. The authors propose a triplet loss with negative mining to train the embedding, which is novel for real-valued time-series. This method is experimentally validated on a classification and a regression task.

General opinion:

* Pros:
    * Good writing
    * Detailed appendix with experimental hyperparameters, so that the results are pretty reproducible.
    * For classification and regression, the proposed method reaches results close to the state-of-the-art.
* Cons:
    * The authors state that the embedding is unsupervised, but in Appendix C they acknowledge that it is trained with early-stopping based on the final classification accuracy, thereby relying on an implicit supervision.
    * This method does not improve over the state-of-the-art on time-series classification, even though it is its natural purpose
    * The experimental validation of the proposed method is weak (cf detailed method).
    * I have some concerns at the conceptual level (cf detailed questions).

Taking into consideration these aspects, I tend to vote for a weak rejection.

Detailed questions:
- On a conceptual level, the ideas underlying the use of a triplet loss explained in the 3rd paragraph of section 2 seems a bit incomplete to me. On the one hand, the authors state that the embedding of a sub-series should be close to the embedding of the series. On the other hand, they also state that this embedding should be far from the embedding of a randomly sampled sub-series, possibly in the same long time-series. This seems contradictory, because if they belong to the same global time-series, they are both sub-series of the global time-series and therefore should be close. Also, the fact that no scale is taken into account when defining sub-series seems quite irrealistic.
- Why is using a *causal* embedding important for classification purposes?
- Experimentally, what are the results when the number of negative samples, K, varies? Experiments have been performed with this parameter varying as an ensemble is taken. It is a pity that the importance of this value is not reported, as it would have provided an intuition on its importance.
- The runtimes reported in Table 1 are a bit strange. Why does the runtime of the raw values vary so much (x30) when moving from daily to quarterly predictions, while the runtime of the representations diminishes (/3)?

---

### Decision · Program_Chairs · 2019-04-11
**Acceptance Decision**

**Decision:**

Accept

**Comment:**


The paper presents some reasonable experiments and approaches for unsupervised time series. However as mentioned by R2 there is several issues. The paper also overclaims  a bit some of the novelty. As noted by  R1 and the metareviewer there is other works using triplet loss for timeseries, a relatively common approach in temporal dataset (e.g. video, audio), the popular causal convolution structure from wavenet is also quite well known, contributions should be more clear.